# Genome-Wide Identification of RNA Editing Sites Affecting Intramuscular Fat in Pigs

**DOI:** 10.3390/ani10091616

**Published:** 2020-09-10

**Authors:** Ligang Wang, Jingna Li, Xinhua Hou, Hua Yan, Longchao Zhang, Xin Liu, Hongmei Gao, Fuping Zhao, Lixian Wang

**Affiliations:** Institute of Animal Science, Chinese Academy of Agricultural Sciences, Beijing 100193, China; ligwang@126.com (L.W.); 82101182354@caas.cn (J.L.); 7hxh73@163.com (X.H.); zcyyh@126.com (H.Y.); zhlchias@163.com (L.Z.); firstliuxin@163.com (X.L.); gaohongmei_123@126.com (H.G.); zhaofuping@caas.cn (F.Z.)

**Keywords:** high-throughput sequencing, RNA editing, muscle growth, intramuscular fat, swine

## Abstract

**Simple Summary:**

In order to investigate RNA editing sites affecting IMF (which is associated with pork quality and human insulin resistance.), we analyzed the transcriptome and genome sequencing data of a high- and low- groups composed of full-sib pairs pig with opposite IMF phenotypes. Finally, a total of 36 nonredundant RNA editing sites in the longissimus dorsi muscle, which may reveal the potential importance of RNA editing in IMF were identified. Four were selected as candidate sites associated with IMF. Our findings provide some new insights of RNA editing function in pig longissimus dorsi muscle.

**Abstract:**

Intramuscular fat (IMF) is essential for improving the palatability and flavor of meat, and it is strongly associated with human insulin resistance. RNA editing is a widespread regulating event in different tissues. Here, we investigated the global RNA editing difference of two groups of pig with different IMF contents to find the potential editing sites affecting IMF. In this research, RES-Scanner and REDItools were used to identify RNA editing sites based on the whole genome and transcriptome sequencing data of the high and low groups composed of three full-sib pairs with opposite IMF phenotypes. A total of 295 RNA editing sites were investigated in the longissimus dorsi muscle, and 90.17% of these sites caused A to G conversion. After annotation, most editing sites were located in noncoding regions (including five sites located on the 3′ UTR regions). Five editing sites (including two sites that could lead to nonsynonymous amino acid changes) were located in the exons of genes. A total of 36 intergroup (high and low IMF) differential RNA editing sites were found in 33 genes. Some candidate editing sites, such as sites in acyl-coenzymeA: cholesterol acyltransferase 1 (*ACAT1*), coatomer protein, subunit alpha (*COPA*), and nuclear receptor coactivator 3 (*NCOA3*), were selected as candidate RNA editing sites associated with IMF. One site located on the 3′ UTR region of growth hormone secretagogue receptor (*GHSR*) may regulate *GHSR* expression by affecting the interaction of miRNA and mRNA. In conclusion, we identified a total of 36 nonredundant RNA editing sites in the longissimus dorsi muscle, which may reveal the potential importance of RNA editing in IMF. Four were selected as candidate sites associated with IMF. Our findings provide some new insights of RNA editing function in pig longissimus dorsi muscle which useful for pig IMF breeding or human insulin resistances research.

## 1. Introduction

RNA editing is the post-transcriptional or co-transcriptional process which could make transcripts more complicated and results in potential functional consequences [1,2]. RNA editing sites have been found in different vertebrates, including humans [3,4,5,6], mice [4], pigs [1,7,8], bovines [9], and chickens [10]. The RNA editing events were usually catalyzed by adenosine deaminase RNA specific (ADAR), and represent the form of A-to-I in mammals [11,12]. RNA editing sites in coding regions of genes can alter the amino acid sequences of proteins [7], and the RNA editing sites in non-coding regions such as 3‘ UTR are involved in editing of miRNA seed regions [13] and target sequences in mRNA [14]. Moreover, RNA editing sites in intergenic region may associated with nuclear retention [15], and in introns may disrupt alternative splicing [16]. In humans, RNA editing is associated with disease [3,4,5]; in animals, RNA editing was found to be associated with embryo and tissue development [1], production traits [17], and Newcastle disease virus [18].

Intramuscular fat (IMF) is highly related to insulin resistance [19] and is highly desirable for meat quality factors such as meat flavor, tenderness, and palatability [20]. Therefore, studies on the mechanisms of IMF development are important for pork quality and human diseases such as type 2 diabetes, obesity, and so on. In pigs, many regulators, such as melatonin [21]; mesenteric estrogen- dependent adipogenesis gene (*Medag*) [22]; lncRNAs, such as MSTRG.604206 (3459 bp to fatty acid synthase (*FASN*)) and MSTRG.426159 (459614 bp to *LOC102160997*) [23]; miR-130a [24]; and miR-125a-5p [25], are significantly related to pig IMF content. These regulatory factors mostly focus on small RNAs, lncRNAs, and protein-coding genes. Studies on the effects of RNA editing sites in porcine IMF development are lacking.

In order to mining the potential effective RNA editing sites on porcine IMF, we first investigate RNA editing sites based on strand-specific RNA sequencing and whole-genome resequencing data of three full-sib paired 240-day-old pigs with different IMF contents. And then, we predicted the potential function of RNA editing sites by binding energy analysis and protein interaction network analysis. And finally, several candidate editing site affecting IMF content. This study not only explores the different RNA editing events between pigs with different IMF contents, but also provides new insight into the functional analysis of RNA editing sites.

## 2. Materials and Methods

### 2.1. Ethics Statements

All animals experiment in our research were carried out under the ethics approval (No. IASCAAS-AE-09) of Animal Ethics Committee of the Institute of Animal Science, Chinese Academy of Agricultural Sciences on 7 September 2018.

### 2.2. Sample Collection and Nucleic Acid Isolation

The longissimus dorsi muscle (10th–11th rib, muscle) were sampling from six 240-day-old Large White X Min pig F2 individuals (3 pairs of full-sib, Table 1). All the tissues were snap frozen in liquid nitrogen and stored in −80 °C refrigerator until use. Total genomic DNA was isolated from the muscle tissues using phenol–chloroform protocol. Total RNA was extracted using TRIzol reagent (Invitrogen, Carlsbad, CA, USA) according to the manufacturer’s instructions. The quantification and quality standards of DNA samples were: (1) optical density (OD) 260/280 ratios of 1.8–2.0, (2) total contents >3 μg. The quality standard of RNA samples mainly was RNA integrity number (RIN) scores ≥7.

### 2.3. Strand-Specific Transcriptome and Whole-Genome Sequencing

Muscle DNA and RNA sequencing libraries were generated using oligo (dT) beads by NEBNext^®^ Ultra™ Directional RNA Library Prep Kit for Illumina^®^ (NEB, Ipswich, MA, USA) and TrSseq Nano DNA HT Sample Preparation Kit (Illumina, San Diego, CA, USA) according to the manufacturer’s recommendations. Then, the sequencing libraries were monitoring by real-time PCR on an Agilent Bioanalyzer 2100 system (Agilent, Lexington, MA, USA) for quantification and quality assessment. At last, libraries constructed above were sequenced on Illumina HiSeq (Illumina, San Diego, CA, USA) platform in Novogene Bioinformatics Technology Cooperation (Beijing, China) by 150 bp paired-end sequencing strategy. All of the clean data have been submitted to the Genome Sequence Archive, with the accession number CRA001645.

### 2.4. RNA Editing Investigation

In order to strengthen robustness of detection, the RNA editing sites were detected using RES-Scanner with the main parameters —ss 1 —uniqTag 1 and default thresholds in user manual [26]. And then, the REDItoolBlatCorrection.py script of REDItools was used for correction [27]. The workflow of RNA editing sites detection were shown as Figure 1. The reference genome, SNP database, and genomic feature files were all download from Ensembl (ftp://ftp.ensembl.org/pub/release-98/fasta/sus_scrofa/). In order to decrease the false positive rate, strict standards were used for RNA editing sites detection as follow: (a) DNA counts ≥10, and homozygous >0.95 (b) counts of RNA reads which differ from DNA genotype ≥3, (c) RNA editing level (editing counts/ total reads of this locus) ≥0.05 and false discovery rate (FDR) <0.05, and (d) candidate editing site is not located within regions of ≥3 residues and ≤6 intronic bases of a splice site.

### 2.5. Validation of Candidate RNA Editing Sites

All of the 295 sequences (50 bp upstream and 50 downstream of each candidate RNA editing sites) were retrieved from the reference genomes. After editing sites altering, all of the sequences were queried against the standard database of expressed sequences tags (ESTs) of swine in NCBI using BLAST algorithm. Five editing sites which could not be found in the DREP (Database of RNA Editing in Pig, http://www.rnanet.org/editing/main.php) and ESTs databases were randomly selected to validate the prediction accuracy of RES-Scanner. DNA and RNA were isolated from the muscle tissues of the same individuals for sequencing. The primers (Appendix A) for sanger sequencing were designed using the software of primer6 and synthesized by Invitrogen Inc. (Shanghai, China). The sanger sequencing was carried by Liuhetong Inc. (Beijing, China). The sites were considered verified if the cDNA sequence was heterozygous while the corresponding DNA sequencing was homozygous.

### 2.6. Annotation of RNA Editing Sites

Annotate variation (ANNOVAR) [28] was used to annotate the genomic features and covered (or nearby) genes of RNA editing sites. Seven types of genomic features which were exonic, ncRNA, downstream, intergenic, UTR, and intronic, and gene Ensembl ID were retrieved. To determine whether the RNA editing sites located in repetitive elements, we used AnnotateTable.py script of REDItools [27].

### 2.7. Functional Enrichment Analysis

We used bioDBnet [29] to convert the Ensembl ID to gene symbol and used Metascape (metascape.org) [30] to peform a functional enrichment analysis based on Gene Ontology (GO) biological processes and Kyoto Encyclopedia of Genes and Genomes (KEGG) pathway terms. GO/KEGG terms with the threshold of FDR <0.05 were considered significantly enriched. For each given gene list, protein-protein interaction enrichment analysis has been carried out with the following databases: BioGrid [31], InWeb_IM [32], OmniPath [33]. The resultant network contains the subset of proteins that form physical interactions with at least one other member in the list. If the network contains between 3 and 500 proteins, the Molecular Complex Detection (MCODE) algorithm [34] has been applied to identify densely connected network components.

### 2.8. Trait Differential RNA Editing Sites Investigation

As described by Peng et al. [35], the sites only edited in high IMF individuals (more than two individuals with editing level >0.15) but not in low IMF individuals were defined as hyper-editing sites. Conversely, the sites only edited in low IMF individuals (more than two individuals with editing level >0.15) but not in high IMF individuals were defined as hypo-editing sites.

### 2.9. Impacts of RNA Editing Events on miRNA-mRNA Interactions

For each hyper- and hypo-editing site located at 3′UTR, we extracted the editing site and the upstream and downstream 25 base pairs (total of 51 bp) from reference genome as an unedited-type (UT) sequence and changed the editing site from A-to-G as an edited-type (ET) sequence. Then, miRANDA [36] was used to calculated the binding energy (with the threshold of −7 kCal/Mol) and to predict the miRNA target sites on UT and ET sequences. The miRNA-mRNA interactions not in UT, but in ET, were defined as interaction losses. The miRNA-mRNA interactions not in ET, but in UT, were defined as interaction gain. The change of binding energy >2 kCal/Mol, were defined as change.

### 2.10. Impacts of RNA Editing Sites on Protein Function

For the A-to-G editing site on the exons, SIFT software [37] was used to predict the on-protein function. Variants with SIFT scores ranged from 0.0 to 0.05 were considered deleterious.

## 3. Results

### 3.1. The Landscape of RNA Editing Sites in Swine Longissimus Dorsi Muscle

To investigate the RNA editing sites at the transcriptome-wide level in longissimus dorsi muscle, DNA sequencing and matched strand-specific RNA sequencing were performed for the six pigs. After quantification and quality filtering, a total of 542 million RNA reads were acquired from the six pigs, with an average mapping rate of 76.7%. A summary of the high-throughput sequencing result is provided in Table 2. After the strand-specific RNA editing analysis, a number of 295 sites (including 266 A-to-G editing sites) were detected (Figure 2A).

Consistent with the previous study in pigs, RNA editing levels of the most editing sites in muscle were low [1,3,8]. Most of the editing sites were under 10.0%, with an average editing level of 21.06% (Figure 2B). Of all detected editing sites 67.46% were located in retained introns (Figure 2C). Also consistent with previous reports, the editing sites were most located in the intronic regions of genes, and these suggested that the RNA editing sites may impact splicing [16]. Only five editing sites were located in the exons, and two of them could cause protein coding changing (Figure 2D). The A-to-G changing of the sites locate on ssc4:14706035 will lead to the glutamine-to-arginine changing of the coding protein of ENSSSCG00000043481, and the sites located on ssc8:84574384 will cause the alanine-to-valine changing of ubiquitin-specific peptidase 38 (USP38). In the results of SIFT prediction, the editing of the sites in USP38 was deleterious to its protein function (*p* < 0.05). As synonymous variants can disrupt transcription, splicing and mRNA stability [38], we used mfold [39] to predict the stability of the RNA structure of the genes containing the synonymous editing sites. Unfortunately, the three synonymous RNA editing sites could not cause any stability changes. Moreover, there are no sites were found to be located in the exon start_0base of the alternative splicing (AS) appearing in the six pigs longissimus dorsi muscle.

### 3.2. Analysis of the Editing Sites Distribution in Chromosome and Repetitive Regions

According to the results of the genomic distribution of the editing sites, the average distances between neighboring editing sites in the longissimus dorsi muscle did not have significant difference across chromosomes (*p* < 0.05). And, this indicated that editing events in the longissimus dorsi muscle have little chromosomal bias (Figure 3A). Consistent with previous studies [1,3,8], 85.76% of the repetitive editing sites were located in short interspersed nuclear elements (SINEs) (Figure 3B). In contrary, only 1.69% of the repetitive editing sites located in long interspersed nuclear elements (LINEs). And included in the SINES, most of the editing sites were located in Pre0_SS elements, with the following PRE1e, PRE1f2, PRE1f, PRE1g, PRE1h, PRE1d, PRE1i, SINE1A_SS, MIR, PRE1c, SINE1_SS, and SINE1D_SS (Figure 3C).

### 3.3. Validation of Candidate RNA Editing Events

According to the in-silico blasting on NCBI, a total number of 235 (79.66%) editing sites were found covering by more than one EST clone (Appendix A), and a total number of 206 (69.83%) editing sites were found in more than one edited EST sequence (Figure 4A). When searching in the database of RNA editing in pig (DREP, http://www.rnanet.org/editing/main.php) [8], about 69.17% (184/266) A-to-G RNA editing sites were found previously reported (Figure 3A). We selected five predicted editing sites that were included neither in the DREP database nor in EST database (totally 35 sites) to perform sanger sequencing. In the results of sanger sequencing, three of the five selected sites could be validated (Figure 4B and Appendix A). In all, maybe 21 sites (35 × 0.6) investigate in our research were first investigated in pig longissimus dorsi muscle.

### 3.4. Functional Enrichment and Interactome Analysis of RNA Editing Sites

After annotation, the RNA editing sites were located in or nearby 254 genes. We converted the ensembl number of these genes to gene official gene symbols, and then submitted genes without synonymous editing sites to Metascape for GO and interactome analysis. Together, we found that genes with editing sites participate in the function of intermuscular fat development, such as MAPK6/MAPK4 signaling. And most of genes with editing sites were associated with muscle development such as myeloid cell development, adherens junction, and skeletal system development. Moreover, genes with editing sites participate in some other biological functions such as the vascular endothelial growth factor receptor signaling pathway, the response to bronchodilators, etc. (Figure 5A). All the genes could interact with each other in four networks, including one large network and three small networks (Figure 5B).

### 3.5. Investigation of Differential RNA Editing Sites in Low and High—IMF Pig Longissimus Dorsi Muscle

A total of 36 hyper- or hypo- editing sites in 33 genes were investigated in all of the editing sites. Most of the sites were also in the introns of genes, and only two sites were located in the 3′UTR region of the genes (Figure 6A). Among the 36 differential RNA editing sites, 31 were hyper-editing sites and only 5 were hypo-editing sites in high-IMF individuals (Figure 6B). Most of the genes with differential editing sites participate in the function of muscle or intermuscular fat development, such as endomembrane system organization, adherens junction, muscle contraction, and the transmembrane receptor protein tyrosine kinase signaling pathway (Figure 6C). Most of the genes are involved in muscle contraction, muscle system process, regulation of muscle system process, etc. (Figure 6D and Appendix A). These genes are located in an independent cluster that contains acyl-coenzymeA: cholesterol acyltransferase 1 (*ACAT1*), solute carrier family 8 member A1 (*SLC8A1*), phosphodiesterase 3A (*PDE3A*), catenin alpha 3 (*CTNNA3)*, tripartite motif containing 72 (*TRIM72*), and follistatin like 1 (*FSTL1*), with the central of growth hormone secretagogue receptor (*GHSR*).

### 3.6. Impacts of RNA Editing Events on Protein Function

The protein–protein interaction (PPI) networks were also constructed using all of the genes containing edited sites in the longissimus dorsi muscle (Figure 7A). As shown in the four MCODE component networks containing the densely-connected proteins, three proteins coded by genes containing differential editing sites could interact with other proteins (Figure 7B) and these three proteins were coatomer protein, subunit alpha (*COPA*), *ACAT1*, and nuclear receptor coactivator 3 (*NCOA3*).

### 3.7. Impacts of RNA Editing Sites on miRNA–mRNA Interactions

When we used miRANDA to calculate the binding ability of miRNA target sites on UT and ET sequences, a total of 4 miRNA–mRNA interaction losses and 2 miRNA-mRNA interaction gains were found in the 3′UTR region of zinc finger protein 543 (*ZNF543*). Five miRNA-mRNA interaction changes were found in the 3′ UTR region of GHSR. Seven miRNA-mRNA interaction gains were found in the 3′UTR region of *GHSR* (Table 3).

## 4. Discussion

In this study, 11 types of RNA editing were detected, including all possible base substitutions as follows: A-to-G, A-to-C, A-to-T, C-to-A, C-to-T, G-to-A, G-to-C, G-to-T, T-to-A, T-to-C, and T-to-G. Overall, the A-to-G substitution was the most common, accounting for up to 90.17% of the identified RNA editing sites. Consistent with previous reports, the average editing level of the A-to-G sites was low overall [1,3,8]. Genes containing editing sites may be involved in various biological functions. However, the genes containing different editing levels in different IMF content groups were mainly involved in the biological function related to myocyte development and lipidosis (Figure 6C and Appendix A). As we know, endomembrane system organization and adherens junction process are the basic processes of myofibrillogenesis. The transmembrane receptor protein tyrosine kinase signaling pathway could regulate energy metabolism and is negatively correlated with obesity [40].

As most of the genes with differential editing sites are associated with the function of muscle, we inferred that most of the sites may have function. As most of the sites were located in the introns, we investigated the alternative splicing (AS) appearing in the six pigs’ longissimus dorsi muscle, and searched the alternative splicing status of all the DE genes. Finally, 23 genes have alternative splicing (Appendix A), but no sites were found to be located in the exon start_0base. And moreover, there are also no sites located on the alternative splicing.

Among the 23 genes, *ACAT1* and *NCOA3* were found in one PPI MCODE network. In this MCODE network, *ACAT1* and *NCOA3* could directly interact with Ras Homolog Family Member A (*RHOA*), and F-Box and WD repeat domain containing 7 (FBXW7) proteins (Figure 6B). FBXW7 was reported to be associated with lipid metabolism [41]. In previous research, *ACAT1* was consider as one of the candidate genes affect ribeye area and backfat thickness [42]. And in the research of Gu et al. [43], stabilization of *FASN* by *ACAT1*-mediated GNPAT acetylation could promote lipid metabolism. Emerging evidence from previous studies using animal models also suggests that the *NCOA3* plays a critical role in lipid metabolism as well as adipogenesis and obesity [44]. So, we and we inferred that editing sites in these two genes may have influences associated with IMF deposition or metabolism.

COPA was also found in another PPI MCODE network containing Chaperonin Containing TCP1 Subunit 6B (CCT6B), Replication Protein A1 (RPA1), DEAD-Box Helicase 6 (DDX6), Proteasome 20S Subunit Beta 2 (PSMB2), DnaJ Heat Shock Protein Family (HSP40) Member A1 (DNAJA1), Cyclin Dependent Kinase 2 (CDK2), and Tubulin Alpha 8 (TUBA8). Overexpression of *CCT6B* significantly inhibited fibroblast migration and collagen synthesis (all *p* < 0.05) [45]. *DDX6* plays essential roles in adipogenesis and in the alternative splicing of peroxisome proliferator activated receptor gamma (*PPARG*) and Lipin 1 (*Lpin1*) [46]. *PSMB2* can be expressed in human brown adipose tissue [47]. *HSP40* plays a vital role to obesity-induced insulin resistance and type 2 diabetes (T2D) [48]. Decreased muscle protein degradation may be regulated though *DNAJA1* pathway in swine [49]. m^6^A-dependent CDK2 expression, mediated by FTO Alpha-Ketoglutarate Dependent Dioxygenase (*FTO*) and YTH N6-Methyladenosine RNA Binding Protein 2 (*YTHDF2*), contributes to adipogenesis inhibition [50]. In Martínez-Montes [17], the *COPA*: g.48255C>T polymorphisms showed effects on backfat thickness. As most of the protein in this MCODE network is associated with adipogenesis, we inferred that *COPA* may be a candidate gene affecting IMF content. And more research, such as RNA pulldown and western blotting experiment could be performed to validate the editing events’effect on interacted proteins.

As the differences in the expression levels of the transcripts in *ZNF543* were minimal between the high- and low-IMF groups, we inferred that although the editing sites in the 3′ UTR region of *ZNF543* have miRNA–mRNA interaction gain/loss or change, the role of this interaction may have no effect on *ZNF543* expression. There are possible post-translational effects that might justify an effect of *ZNF543* on IMF. One transcript, named *GHSR-202*, showed a 10.59-fold change expression difference between the high- and low-IMF groups. We searched all of the potential interactions with miRNAs in PubMed. ssc-miR-15b, ssc-miR-16, ssc-miR-103, and ssc-miR-221-3p were reported to have different expressions in the myofiber between lean and obese pig breeds [51]. ssc-miR-16 was also reported to be upregulated in the muscle of obese minipigs [52]. ssc-miR-103 and ssc-miR-221-3p are associated with meat quality [53]. Ghrelin exerts direct peripheral effects on lipid metabolism, including increasing white adipose tissue mass, stimulating lipogenesis in the liver, and taste sensitivity modulation. Ghrelin and its receptor *GHSR* are involved in obesity [54,55,56]. Combined with the results of *GHSR* being the center of the GO cluster, we speculated that *GHSR* may play a crucial part in IMF regulation. The editing site in the 3′ UTR of *GHSR* may contribute to the lipogenesis by regulating the miRNA of ssc-miR-15b, ssc-miR-16, ssc-miR-103, or ssc-miR-221-3p. To determine the target miRNA, further research, such as dual luciferase reporter assay should be performed.

## 5. Conclusions

In this study, a total of 295 RNA editing sites were identified using strand-specific RNA and whole-genome DNA sequencing data in pig longissimus dorsi muscle. The functional analysis of the genes with hyper- and hypo-editing sites revealed the potentially functional importance of RNA editing in porcine IMF regulation. Some candidate editing sites, such as in *ACAT1*, *NCOA3*, *COPA*, and especially *GHSR*, should be deeply studied in future research. More research such as RNA pulldown, western blotting and dual luciferase reporter assay should be done to explore the relationship between these candidate editing sites and interacted genes and proteins. Our findings provide some new RNA editing sites and some new insights of RNA editing function which useful for pig IMF breeding or human insulin resistances research.

## Figures and Tables

**Figure 1 animals-10-01616-f001:**
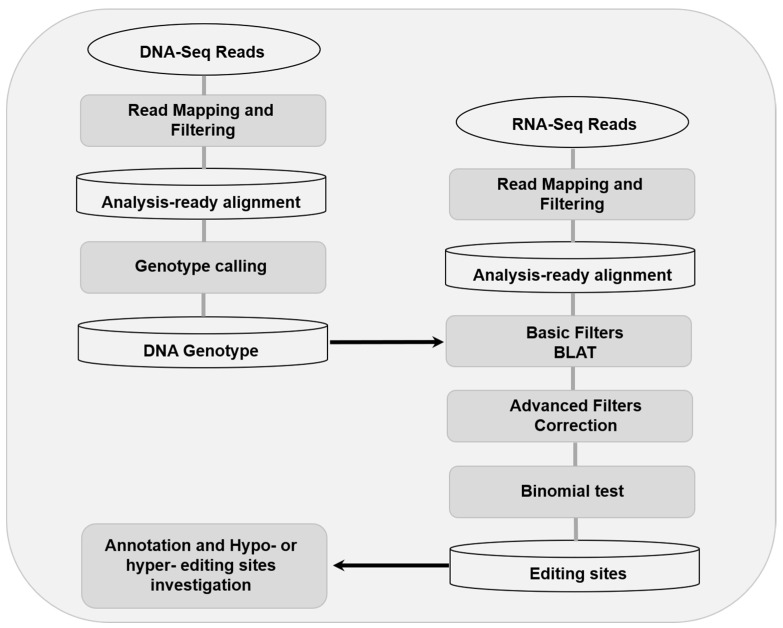
Overview of the workflow of RNA editing sites detection.

**Figure 2 animals-10-01616-f002:**
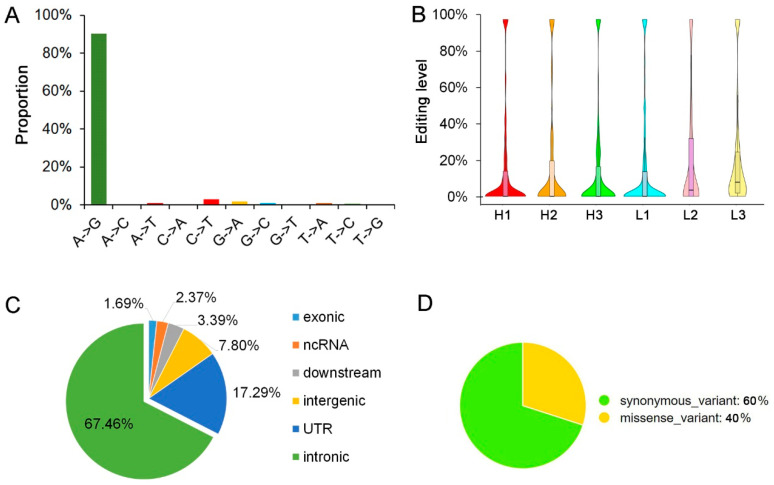
Landscape of the pig longissimus dorsi muscle editome. (**A**) Distribution statistic of detected RNA editing types. (**B**) The violin plot of editing levels of all of the editing sites. (**C**) Genomic features of RNA editing sites in or nearby genes. (**D**) Proportion of synonymous and missense exonic editing sites.

**Figure 3 animals-10-01616-f003:**
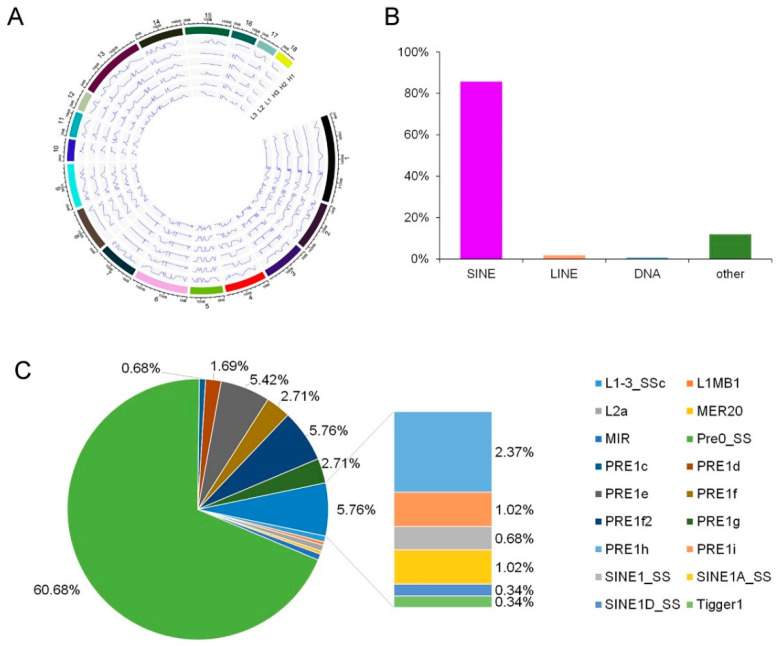
Distribution of RNA editing sites across the genome. (**A**) Distribution of RNA editing sites across chromosomes. The peaks showed the appearance of editing, and the valleys showed the disappearance of editing sites. (**B**) Distribution of RNA editing sites across repetitive classes. SINEs, short interspersed nuclear elements; LINEs, long interspersed nuclear elements. (**C**) Distribution of RNA editing sites across repetitive elements.

**Figure 4 animals-10-01616-f004:**
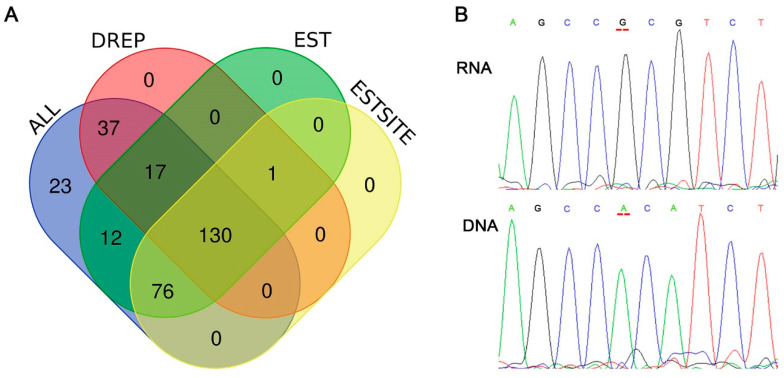
Validation of RNA editing sites. (**A**) Venn diagram of the RNA editing sites searching in other databases. EST, Expression site tags; DREP, Database of RNA Editing in Pig; ALL, all of the 295 editing sites detected in this research; ESTSITE, EST clones with known editing sites. (**B**) Sanger sequencing plot showing different nucleotide (A and G) between genomic DNA and RNA at same position (editing site at Sus scrofa chromosome 13: 111023145-). The sites are highlighted in red lines.

**Figure 5 animals-10-01616-f005:**
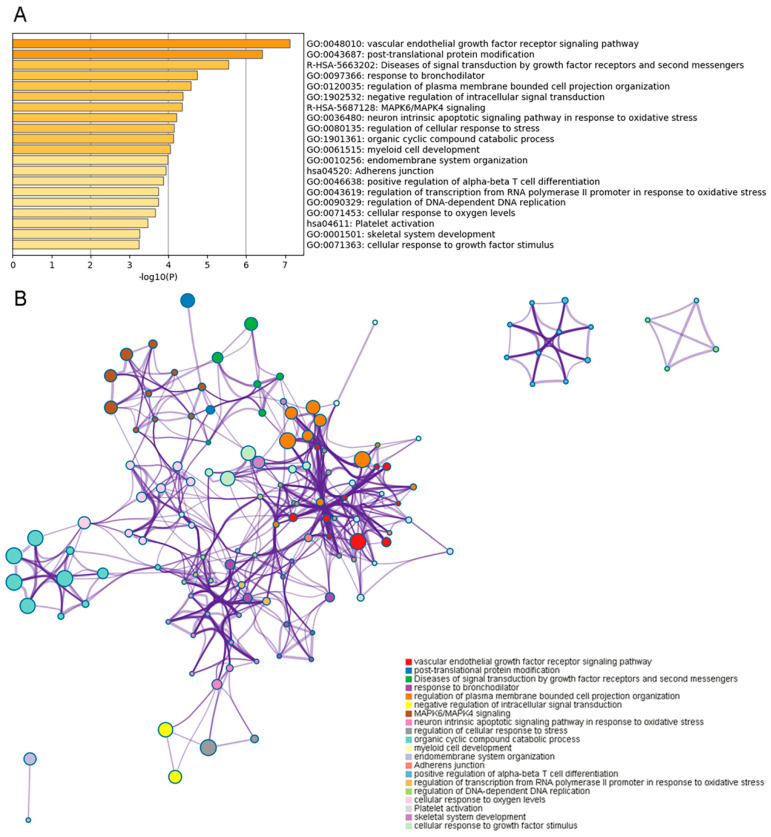
Gene Ontology (GO) cluster and interactome analysis of the genes containing RNA editing. (**A**) Top significant GO biological functions of the genes with RNA editing sites. (**B**) The interactome network of the genes with RNA editing sites. The interaction represents GO-based functional relationship. Different biological functions are represented in different colors.

**Figure 6 animals-10-01616-f006:**
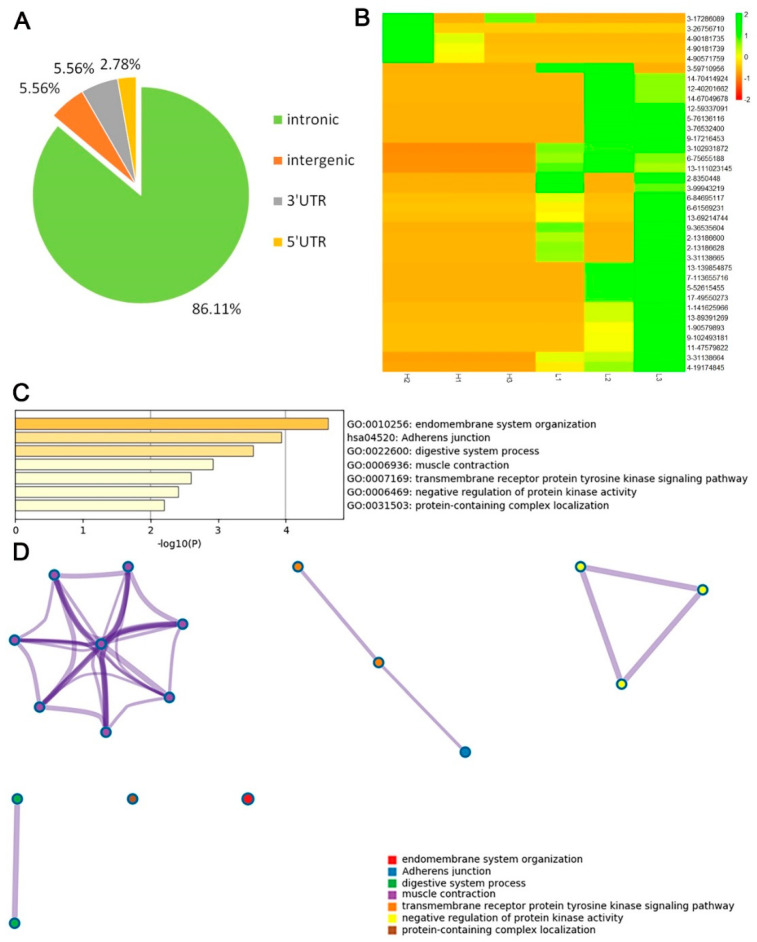
Distribution and function analysis of genes containing differential RNA editing sites. (**A**) Genomic features of hyper- and hypo-editing sites in genes. (**B**) Cluster heatmap of differential editing sites. Different color represents different editing level. (**C**) Most significant GO biological function of the genes with hyper- or hypo-editing sites. (**D**) The interactome network of the genes with hyper- or hypo-editing sites. Different biological functions are represented in different colors.

**Figure 7 animals-10-01616-f007:**
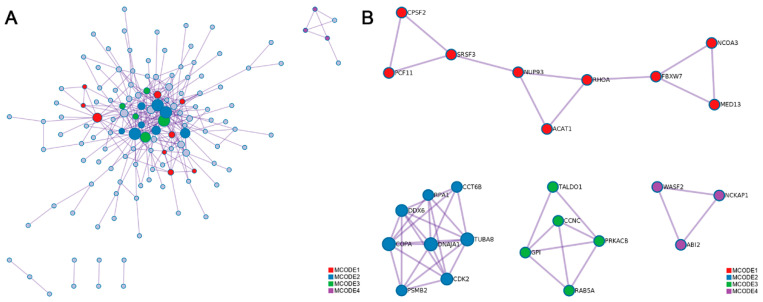
Protein–protein interaction (PPI) network of genes with differential RNA editing sites. (**A**) PPI network of all of the coding proteins. (**B**) The independent PPI networks of molecular complex detection (MCODE) components. The proteins in same MCODE represent densely connection. Different MCODE networks were shown in different color.

**Table 1 animals-10-01616-t001:** Pedigree and phenotype information of selected pigs.

ID	Group	Group ID	Father ID	Mother ID	IMF ^a^ Content (%)
19803	L	L1	721205	723604	0.9
1015105	L	L2	706601	706204	1.41
1119609	L	L3	700105	709602	1.08
19809	H	H1	721205	723604	5.56
1015103	H	H2	706601	706204	5.94
1119605	H	H3	700105	709602	7.51

^a^ IMF: Intramuscular fat.

**Table 2 animals-10-01616-t002:** Summary of the high-throughput sequencing.

Individual	RNA-Seq		DNA-Seq		
	Total Reads	Mapped Rate	Total Reads	Mapping Rate	Coverage ^1^
H1	86,985,200	77.1%	423,325,180	87.2%	81.3%
H2	80,100,158	77.7%	476,288,926	90.4%	84.3%
H3	81,579,006	75.3%	435,526,280	88.9%	82.1%
L1	115,906,328	75.8%	500,321,520	87.2%	85.9%
L2	91,861,266	77.3%	476,238,994	89.4%	84.2%
L3	86,400,546	77.25%	456,625,288	86.8%	81.9%

^1^ The coverage was estimated based autosomal and X chromosomes.

**Table 3 animals-10-01616-t003:** miRNAs ^1^ that represent differential binding ability to different type of editing sites.

Gene	miRNA ID	Type
*GHSR* ^2^	ssc-miR-15b	change
	ssc-miR-20a-3p	gain
	ssc-miR-216	gain
	ssc-miR-217	gain
	ssc-miR-103	change
	ssc-miR-107	change
	ssc-miR-16	gain
	ssc-miR-221-3p	loss
	ssc-miR-503	change
	ssc-miR-497	gain
	ssc-miR-222	loss
	ssc-miR-4339	change
	ssc-miR-187	gain
	ssc-miR-2483	gain
	ssc-miR-9858-5p	loss
*ZNF543* ^3^	ssc-miR-145-3p	loss
	ssc-miR-30e-3p	loss
	ssc-miR-664-3p	loss
	ssc-miR-9849-5p	gain
	ssc-miR-9861-5p	loss
	ssc-miR-10383	gain

^1^ miRNAs, microRNAs; ^2^
*GHSR*, growth hormone secretagogue receptor; ^3^
*ZNF543*, Zinc finger protein 543.

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
