# Peer review of "Genome-Wide Identification of RNA Editing Sites Affecting Intramuscular Fat in Pigs"

_animals, 2020, doi:10.3390/ani10091616_

Round 1

Reviewer 1 Report

## Overall comment for both editor and authors 

The manuscript addresses an essentially high-throughput bioinformatics assessment of RNA editing sites in muscle cells' transcripts from two different swine groups: low and high intramuscular fat (IMF) content. Despite being a basic/fundamental study, the authors make clear a foreseeable economic importance of such research on a production domestic animal. Some muscle-related findings might be of impact if further pursued towards thorough reductionist molecular characterizations on target transcripts, which authors claimed interest on doing so at the end of the article (Conclusion section). There are a number of minor/moderate revisions required in order to improve readability and clarify methodologies/results. Please see my itemized comments below.

## Revisions required 

# Abstract
line 29: I would replace "extrons" by "exons" herein and in all occurrences throughout the manuscript.

# Introduction
lines 46-49: The sentence within these lines must be ameliorated. Try to focus on regulatory events which non-coding regions (including both ncRNAs and UTRs from mRNAs) may act upon. The points briefly described are not necessarily wrong, but are quite vague and, therefore, misleading/confusing.
line 56: These MSTRG.604206 and MSTRG.426159 accession numbers doesn't mean anything, since it is automatically generated by the StringTie transcriptome assembler. Please check whether the reference has submitted their sequences and genomic coordinates to any curated database and provide their respective proper/official gene ID.
line 57: Wouldn't the authors mean "regulatory" rather than "regulate"?
lines 60-65: This whole paragraph should be revised for a proper writing style. Moreover, it does not properly conclude the introduction showing to readers the overall importance of the findings.

# Methods
lines 88-89: Please specify the sequencing strategy (single read or paired-end) as well as the reads' length. Also please submit all raw data (fastq files) to either SRA or ENA repositories and provide the readers with samples' accession numbers.
line 91: replace "was" by "were", and please pay attention for fixing all similar verbal agreement errors that exist throughout the manuscript.
lines 91-98: I missed the description of REDITools usage, since it was mentioned in the abstract that both RES-Scanner and REDItools were used. A combination of results from both tools would strengthen robustness of the findings. The authors should consider doing so or reasonably explain why they haven't performed that (and adjust abstract text).
line 130: Please specify the free-energy threshold used in miRanda executions.
lines 131-132: Wouldn't the authors mean "but not in" rather than "but in"?

# Results
line 152: please cite the previous study.
lines 158-159: please fix "site located" ob both occurrences. Also, this "changing of a novel gene" statement is ambiguous. Please properly adjust it to make it clearer.
line 164: replace "chromosome" by "chromosomes".
lines 164-165: Figure 2A needs to be better explained in both text body and figure legend. What would be the metric for assessing chromosomal bias in that case? The legend must explain what peaks/valleys mean.
line 177: Please clarify/specify which sequence database(s) the Blast comparison(s) was/were performed against. On Fig.3A legend, it is quite hard to understand what ALL and ESTSITE really mean.
On Fig.3B, by what I've checked through ensembl genome browser, the target editing site position (chromosome 13: 111023145+) appears to be on the bottom DNA strand, therefore, minus instead of plus. Moreover, GHSR-203 gene is on the reverse orientation (placed on minus strand). Please double check it and adjust accordingly.
line 194: I did not get the reason of the word "Statistically". Where are the statistical metrics for this statement?
lines 195-196: This sentence must be improved for a better clarification to readers.
lines 200-202: Which kind of interactions are those in Fig.4B (PPI or simply GO-based functional relationship)? This must be clarified to readers.
line 208: I got a bit confused with what "Trait differential RNA editing sites investigation" could be meaning. Please ameliorate this topic title.
On Fig.5B, the color key metric for the heatmap must be explained in the legend.
line 231: Please cite MCODE (Bader & Hogue, 2003).
line 232: Replace "interacted" by "interact"
lines 241-243: This sentence needs several english corrections.
line 244: There's a disagreement between text and table. There are only 7 gains for GHSR depicted on Table 3.
Still on Table 3, replace "lose" by "loss".

# Discussion
Please refer to "alternative splicing" rather than "alternative splice(s)", on all its occurrences.
On Table S4, there are some retained introns alternative splicing category. I wonder if the editing sites would be located on those. The authors should check (or double check) that in order to inform readers about such findings that might be of interest.

# Conclusions
line 303: findings can't investigate anything. Please adjust this sentence.

Author Response

Reviewer 1:

The manuscript addresses an essentially high-throughput bioinformatics assessment of RNA editing sites in muscle cells' transcripts from two different swine groups: low and high intramuscular fat (IMF) content. Despite being a basic/fundamental study, the authors make clear a foreseeable economic importance of such research on a production domestic animal. Some muscle-related findings might be of impact if further pursued towards thorough reductionist molecular characterizations on target transcripts, which authors claimed interest on doing so at the end of the article (Conclusion section). There are a number of minor/moderate revisions required in order to improve readability and clarify methodologies/results. Please see my itemized comments below.

## Revisions required 

# Abstract
line 29: I would replace "extrons" by "exons" herein and in all occurrences throughout the manuscript.

Response: Thank you very much for the serious and responsible review and recognition of our manuscript. We have replaced "extrons" by "exons" in all occurrences throughout the manuscript (Line 29, and Line 152).

# Introduction
lines 46-49: The sentence within these lines must be ameliorated. Try to focus on regulatory events which non-coding regions (including both ncRNAs and UTRs from mRNAs) may act upon. The points briefly described are not necessarily wrong, but are quite vague and, therefore, misleading/confusing.

Response: Follow your suggestion, the potential function of RNA editing sites in ncRNAs, UTRs from mRNAs, intergenic regions, and introns were described separately (Line 48-50).

line 56: These MSTRG.604206 and MSTRG.426159 accession numbers doesn't mean anything, since it is automatically generated by the StringTie transcriptome assembler. Please check whether the reference has submitted their sequences and genomic coordinates to any curated database and provide their respective proper/official gene ID.

Response: We have double checked the reference (including supporting information) and found none proper / official gene ID of these two lncRNAs (MSTRG.604206 and MSTRG.426159). Instead, we added the genomic location information of these two lncRNAs (Line 57-58).

line 57: Wouldn't the authors mean "regulatory" rather than "regulate"?

Response: Yes, we mean "regulatory" rather than "regulate", and we have revised this word in the manuscript (Line 59)

lines 60-65: This whole paragraph should be revised for a proper writing style. Moreover, it does not properly conclude the introduction showing to readers the overall importance of the findings.

Response: Follow your suggestion, we have deleted the first sentence to strengthen the transition with the previous paragraph. Moreover, sentences which described our research and pointed the importance of our findings were also added (Line 62-70).

# Methods
lines 88-89: Please specify the sequencing strategy (single read or paired-end) as well as the reads' length. Also please submit all raw data (fastq files) to either SRA or ENA repositories and provide the readers with samples' accession numbers.

Response: In our research, 150 bp paired-end reads sequencing strategy were used. All of the clean data have been submitted to the Genome Sequence Archive, with the accession number CRA001645. (Line 93-95)

line 91: replace "was" by "were", and please pay attention for fixing all similar verbal agreement errors that exist throughout the manuscript.

Response: We have replaced "was" by "were" here (Line 97) and all other similar occurrences throughout the manuscript (Line170, and Line 272)

lines 91-98: I missed the description of REDITools usage, since it was mentioned in the abstract that both RES-Scanner and REDItools were used. A combination of results from both tools would strengthen robustness of the findings. The authors should consider doing so or reasonably explain why they haven't performed that (and adjust abstract text).

Response: In order to strengthen robustness of detection, RES-Scanner was used to the RNA editings detection first in our research, and then, the REDItoolBlatCorrection.py script of REDItools was used for correction. We have added this in the paragraph (Line 97-100)

line 130: Please specify the free-energy threshold used in miRanda executions.

Response: The free-energy threshold used in miRanda executions was -7 kCal/Mol, and we have added this in the manuscript (Line 146)

lines 131-132: Wouldn't the authors mean "but not in" rather than "but in"?

Response: Yes, we have corrected this in the manuscript (Line 147-148)

# Results
line 152: please cite the previous study.

Response: Reference [1, 3, 8] were cited here (Line 170)

lines 158-159: please fix "site located" ob both occurrences. Also, this "changing of a novel gene" statement is ambiguous. Please properly adjust it to make it clearer.

Response:  The RNA editing will cause the coding protein change. In order to make the sentence clearer, we have added the ensembl ID of the novel gene, and fixed the two sentences together. (Line 176-177)

line 164: replace "chromosome" by "chromosomes".

Response: Done. (Line 185)

lines 164-165: Figure 2A needs to be better explained in both text body and figure legend. What would be the metric for assessing chromosomal bias in that case? The legend must explain what peaks/valleys mean.

Response: As the average distance between two neighboring editing sites did not has significant differences (P < 0.05) in each chromosome, we conclude that RNA editing in longissimus dorsi muscle has none chromosomal bias (Line 186-188). The peaks showed the appearance of editing, and the valleys showed none editing sites appearance. And we have added this in the manuscript (Line 197-198)

line 177: Please clarify/specify which sequence database(s) the Blast comparison(s) was/were performed against. On Fig.3A legend, it is quite hard to understand what ALL and ESTSITE really mean.

Response:  The standard database of expressed sequences tags of swine in NCBI was used to perform the Blast comparison in our research (Line 112-113). And on Fig.3A legend, ALL stand for all of the 295 editing sites detected in our research, and ESTSITE stands for EST clones which contain known editing sites. And we have clarified this in the paper (Line 214-215).

On Fig.3B, by what I've checked through ensembl genome browser, the target editing site position (chromosome 13: 111023145+) appears to be on the bottom DNA strand, therefore, minus instead of plus. Moreover, GHSR-203 gene is on the reverse orientation (placed on minus strand). Please double check it and adjust accordingly.

Response: After double checking, we found that the editing site was on minus strand and on the 3’ URT region of GHSR-203. The mistake has been corrected in the manuscript (Line 217)

line 194: I did not get the reason of the word "Statistically". Where are the statistical metrics for this statement?

Response: After annotation, the RNA editing sites were located in or nearby 254 genes. And we have corrected this in the manuscript. (Line 219-220)

lines 195-196: This sentence must be improved for a better clarification to readers

Response: we have clarified GO terms to biological function separately, such as MAPK6/MAPK4 signaling to intermuscular fat development, myeloid cell development, adherence junction, and skeletal system development to muscle development, and so on (Line 223-225)

lines 200-202: Which kind of interactions are those in Fig.4B (PPI or simply GO-based functional relationship)? This must be clarified to readers.

Response: The interaction showed in fig.4B was the GO-based functional relationship. And we have clarified this in the figure legend (Line 228-229)

line 208: I got a bit confused with what "Trait differential RNA editing sites investigation" could be meaning. Please ameliorate this topic title.

Response: The topic title has been changed to “Investigation of differential RNA editing sites in low and high – IMF pig longissimus dorsi muscle. (Line 232-233)

On Fig.5B, the color key metric for the heatmap must be explained in the legend.

Response: On Fig.5B, different color represents different editing level. And we have added this in the manuscript. (Line 254)

line 231: Please cite MCODE (Bader & Hogue, 2003).

Response: Done. (reference [34], Line 136)

line 232: Replace "interacted" by "interact"

Response: Done. (Line 261)

lines 241-243: This sentence needs several english corrections.

Response: We have corrected the plural form of loss and gain, and replaced “was” to “were” in this sentence. (Line 271-272).

line 244: There's a disagreement between text and table. There are only 7 gains for GHSR depicted on Table 3.

Response: There are only 7 miRNA–mRNA interaction gains were found in the 3’ UTR region of GHSR. And we have corrected this in the manuscript. (Line 273-274)

Still on Table 3, replace "lose" by "loss".

Response: Done. (Line 275 Table 3.)

# Discussion
Please refer to "alternative splicing" rather than "alternative splice(s)", on all its occurrences.

Response: We have replaced " alternative splice(s) " by " alternative splicing " on all its occurrences throughout the manuscript. (Line290-291,and Line 350)

On Table S4, there are some retained introns alternative splicing category. I wonder if the editing sites would be located on those. The authors should check (or double check) that in order to inform readers about such findings that might be of interest.

Response: We have double checked the results, unfortunately, there is no site located on the alternative splicing. Anyway, we describe this in the manuscript. (Line 292-293)

# Conclusions
line 303: findings can't investigate anything. Please adjust this sentence.

Response: We have replaced “investigate” by provide, and reorganized the sentence (Line 344-347).

Reviewer 2 Report

Overall, the manuscript is well written with figures supporting their results. I am not an expert in RNA editing and sequencing so I would like to clarify that I cannot provide an in-depth review regarding those specific results.

Here are a few of areas for improvement:

1. There are a ton of grammatical and spelling errors in the Abstract and Introduction section and needs to be completely re-written.

2. The significance of their research is understated, especially in the Discussion. Please emphasize what is the importance of these genes they found and any future impact as well as the next steps of this study.

3. A summary diagram showing their experimental workflow from start to finish is essential.

Author Response

Reviewer 2:

Overall, the manuscript is well written with figures supporting their results. I am not an expert in RNA editing and sequencing so I would like to clarify that I cannot provide an in-depth review regarding those specific results.

Here are a few of areas for improvement:

  1. There are a ton of grammatical and spelling errors in the Abstract and Introduction section and needs to be completely re-written.

Response: Thank you very much for the serious and responsible review and recognition of our manuscript. We have corrected the grammatical and spelling errors in the abstract (Line 13-14, Line 29, and Line 38) and almost re-written the introduction section (Line 42-70).

  1. The significance of their research is understated, especially in the Discussion. Please emphasize what is the importance of these genes they found and any future impact as well as the next steps of this study.

Response: We have added some references [41-44] to emphasize the importance of the selected genes (Line 297-303). The future impact on IMF breeding and human insulin resistance research, and the next steps of our study were also added in both discussion and conclusion part (Line317-319, Line335-336, and Line 342-247).

  1. A summary diagram showing their experimental workflow from start to finish is essential.

Response: Follow the reviewer’s suggestion, we have added a diagram to show our experimental workflow (Figure 1, Line 108).

Reviewer 3 Report

I've found this piece of work very well articulated and discussed. Just 2 minor issues:

Line 88-89 Please include the sequencing length and method (e.g. Pair-end 150bp )

Line 117 please change “covert the Ensembl” to “convert”

Author Response

Reviewer 3:

I've found this piece of work very well articulated and discussed. Just 2 minor issues:

Line 88-89 Please include the sequencing length and method (e.g. Pair-end 150bp)

Response: We have added the sequencing strategy (paired-end 150bp) in this place. (Line 93-94)

Line 117 please change “covert the Ensembl” to “convert”.

Response: Done. (Line 128)

Reviewer 4 Report

The authors of the manuscript entitled “Genome-wide identification of RNA editing sites affecting intramuscular fat in pigs” performed a prospection for RNA editing sites that might be associated with intramuscular fat content in pigs. The methodology applied on the current manuscript scientifically sounds and the results presented are relevant for the current literature context regarding the biological processes associated with intramuscular fat composition in pigs. There are some minor comments that I would like to list.

My major concern is about the English. The manuscript must pass by a deep review regarding the writing. There are several grammar errors (some are listed below) and in some moments the text is hard to read due to the structure of the sentences. Additionally, why the authors decided to remove the synonymous variants from the functional analysis? Synonymous variants can result in functional impacts due to changes in the codon usage and other processes. The authors also must provide all the details about the functional analyses performed in the manuscript. For example, the PPI network analysis is not described in the material and methods. Please, review the manuscript and provide all the proper details. Regarding the detection of RNA editing sites, the authors should provide the parameters used and not only refer to the original reference. Finally, the discussion section is very short. The authors could provide more details about the candidate genes and to provide a better association among the candidate genes and the intramuscular fat composition. For example, a more detailed discussion about ACAT1 and NCOA3.

Zeng, Z., & Bromberg, Y. (2019). Predicting functional effects of synonymous variants: a systematic review and perspectives. Frontiers in genetics10, 914.

Minor comments:

Line 13: replace “we analysis” to “we analyzed the”

Line 14: replace group to groups

Line 29: extrons?

Line 42: replace posttranscriptional to post-transcriptional. A quick search on google scholar shows that the second option is ~8x more commonly used.

Line 65: Replace “function” to “functional”

Line 120: the q-value was calculated using which multiple testing correction? FDR, Bonferroni?

Line 183-185: Can the authors really say that 21 sites out of the initial 35 not previously described are not a false-positive? The authors are based this in the results of a Sanger sequencing after randomly sample 5 variant sites. This frequency might change either to lower or higher false-positive frequencies depending of the number of sites sampled.

Line 194: “in or”. Is there something missing here?

Line 195: Replace ensemble to ensembl.

Line 264: Add a space between “ACAT1” and “and”.

Lines 285-286: Are there possible post-translational effects that might justify an effect of ZNF543 on IMF? Consequently, these effects will not change the expression values.

Lines 299-305: The conclusion should provide a more detailed description about the future perspectives and applications pf the current results, as well as, what should be performed in order to validade the results.

Author Response

Reviewer 4:

The authors of the manuscript entitled “Genome-wide identification of RNA editing sites affecting intramuscular fat in pigs” performed a prospection for RNA editing sites that might be associated with intramuscular fat content in pigs. The methodology applied on the current manuscript scientifically sounds and the results presented are relevant for the current literature context regarding the biological processes associated with intramuscular fat composition in pigs. There are some minor comments that I would like to list.

My major concern is about the English. The manuscript must pass by a deep review regarding the writing. There are several grammar errors (some are listed below) and in some moments the text is hard to read due to the structure of the sentences. Additionally, why the authors decided to remove the synonymous variants from the functional analysis? Synonymous variants can result in functional impacts due to changes in the codon usage and other processes. The authors also must provide all the details about the functional analyses performed in the manuscript. For example, the PPI network analysis is not described in the material and methods. Please, review the manuscript and provide all the proper details. Regarding the detection of RNA editing sites, the authors should provide the parameters used and not only refer to the original reference. Finally, the discussion section is very short. The authors could provide more details about the candidate genes and to provide a better association among the candidate genes and the intramuscular fat composition. For example, a more detailed discussion about ACAT1 and NCOA3.

Zeng, Z., & Bromberg, Y. (2019). Predicting functional effects of synonymous variants: a systematic review and perspectives. Frontiers in genetics, 10, 914.

Response:

Thank you very much for the serious and responsible review and recognition of our manuscript. Although the article has undergone English language editing by MDPI (NO.209458), some writing problem still existed. Follow the suggestion of reviewers, we have checked and edited English in the manuscript thoroughly and invited one native speaker to improve our language.

As described in the reference (Zeng, et al., 2019), synonymous variants can disrupt transcription, splicing and mRNA stability. And follow the reviewer`s suggestion, we used mfold to predict the stability of the RNA structure, and found that the three synonymous RNA editing events could not cause any stability changes. Moreover, there were also no sites found to be located in the exon start_0base of the alternative splicing appearing in the six pigs longissimus dorsi muscle. And we have added these results in the manuscript. (Line 179-184)

In our research, for each given gene list, protein-protein interaction enrichment (PPI) analysis has been carried out with the following databases: BioGrid, InWeb_IM, and OmniPath. The resultant network contains the subset of proteins that form physical interactions with at least one other member in the list. If the network contains between 3 and 500 proteins, the Molecular Complex Detection (MCODE) algorithm has been applied to identify densely connected network components. And we have added this in the manuscript. (Line 131-136)

In our research, the RNA editing sites were detected using RES-Scanner with the main parameters --ss 1 --uniqTag 1 and default thresholds. And then, the REDItoolBlatCorrection.py script of REDItools was used for correction. And we have added this in the manuscript. (Line 97-100 )

In some research (Silva-Vignato B, et al., 2019), ACAT1 was consider as one of the candidate genes affect ribeye area and backfat thickness. And in the research of Gu et al. (2020), stabilization of FASN by ACAT1-mediated GNPAT acetylation could promote lipid metabolism. Emerging evidence from previous studies using animal models (Yu et al. 2015) also suggests that the NCOA3 plays a critical role in lipid metabolism as well as adipogenesis and obesity. And combined with the analysis in our article, we inferred that the editing sites in these two genes may have influences associated with IMF deposition or metabolism. And we have added this in the manuscript. (Line 297-303)

Minor comments:

Line 13: replace “we analysis” to “we analyzed the”

Response: Done. (Line 13)

Line 14: replace group to groups

Response: Done. (Line14)

Line 29: extrons?

Response: Yes, and we have replaced "extrons" by "exons" in all occurrences throughout the manuscript. (Line 29, and Line 152)

Line 42: replace posttranscriptional to post-transcriptional. A quick search on google scholar shows that the second option is ~8x more commonly used.

Response: We have replaced posttranscriptional to post-transcriptional. (Line42), and we did not find any related option you described in our paper.

Line 65: Replace “function” to “functional”

Response: This sentence has been rewritten. (Line 66-70)

Line 120: the q-value was calculated using which multiple testing correction? FDR, Bonferroni?

Response: In our research, the q-value was calculated using FDR. And we revised this in the manuscript. (Line 131)

Line 183-185: Can the authors really say that 21 sites out of the initial 35 not previously described are not a false-positive? The authors are based this in the results of a Sanger sequencing after randomly sample 5 variant sites. This frequency might change either to lower or higher false-positive frequencies depending of the number of sites sampled.

Response: As the editing sites sampling number is small, the statement that “21 sites investigate in our research were first investigated in pig longissimus dorsi muscle” is not rigorous. We have replaced the word “about” by “maybe” in the article according to the possible false-positive. (Line 209-210)

Line 194: “in or”. Is there something missing here?

Response: The sentence has been revised according to reviewer1’s suggestion. (Line 219-220)

Line 195: Replace ensemble to ensembl.

Response: Done. (Line 220)

Line 264: Add a space between “ACAT1” and “and”.

Response: Done. (Line 294)

Lines 285-286: Are there possible post-translational effects that might justify an effect of ZNF543 on IMF? Consequently, these effects will not change the expression values.

Response: Although the expression values of ZNF53 is not changed, possible post-translational effects that might justify an effect of ZNF543 on IMF still existed. And we have added this comment in the paper. (Line 323-324)

Lines 299-305: The conclusion should provide a more detailed description about the future perspectives and applications pf the current results, as well as, what should be performed in order to validade the results.

Response: In the next step of our research, RNA pulldown and western blotting experiment could be performed to validate the editing events’effect on interacted proteins. The dual luciferase reporter assay could be performed to validate the influence of RNA editing on the binding of miRNAs to mRNAs. And we have added these in both discussion and conclusion part of the manuscript. (Line 317-319, Line335-336 and Line 342-347)